# Recombinant Muscovy Duck Parvovirus Led to Ileac Damage in Muscovy Ducklings

**DOI:** 10.3390/v14071471

**Published:** 2022-07-03

**Authors:** Jiahui He, Yukun Zhang, Zezhong Hu, Luxuan Zhang, Guanming Shao, Zi Xie, Yu Nie, Wenxue Li, Yajuan Li, Liyi Chen, Benli Huang, Fengsheng Chu, Keyu Feng, Wencheng Lin, Hongxin Li, Weiguo Chen, Xinheng Zhang, Qingmei Xie

**Affiliations:** 1Heyuan Branch, Guangdong Provincial Laboratory of Lingnan Modern Agricultural Science and Technology, College of Animal Science, South China Agricultural University, Guangzhou 510642, China; hjh1239417073@163.com (J.H.); ykzhang20@163.com (Y.Z.); 20203144002@stu.scau.edu.cn (Z.H.); sgm19951015@163.com (G.S.); xiezi@scau.edu.cn (Z.X.); nynieyu2022@163.com (Y.N.); 18846921030@163.com (W.L.); lyj119963@163.com (Y.L.); lychen@stu.scau.edu.cn (L.C.); hbl2016310866@163.com (B.H.); cfs13167792558@163.com (F.C.); fky19842004@163.com (K.F.); wenchenglin@scau.edu.cn (W.L.); hxli@scau.edu.cn (H.L.); wgchen@126.com (W.C.); 2Guangdong Engineering Research Center for Vector Vaccine of Animal Virus, Guangzhou 510642, China; 3South China Collaborative Innovation Center for Poultry Disease Control and Product Safety, Guangzhou 510642, China; 4School of Pharmaceutical Science, Sun Yat-sen University, Guangzhou 510006, China; z121727845@126.com; 5Key Laboratory of Animal Health Aquaculture and Environmental Control, Guangzhou 510642, China

**Keywords:** waterfowl parvovirus, inflammation, ileac microflora, oxidative stress, molecular characterization

## Abstract

Waterfowl parvovirus (WPFs) has multiple effects on the intestinal tract, but the effects of recombinant Muscovy duck parvovirus (rMDPV) have not been elucidated. In this study, 48 one-day-old Muscovy ducklings were divided into an infected group and a control group. Plasma and ileal samples were collected from both groups at 2, 4, 6, and 8 days post-infection (dpi), both six ducklings at a time. Next, we analyzed the genomic sequence of the rMDPV strain. Results showed that the ileal villus structure was destroyed seriously at 4, 6, 8 dpi, and the expression of ZO-1, Occludin, and Claudin-1 decreased at 4, 6 dpi; 4, 6, 8 dpi; and 2, 6 dpi, respectively. Intestinal cytokines IFN-α, IL-1β and IL-6 increased at 6 dpi; 8 dpi; and 6, 8 dpi, respectively, whereas IL-2 decreased at 6, 8 dpi. The diversity of ileal flora increased significantly at 4 dpi and decreased at 8 dpi. The bacteria Ochrobactrum and Enterococcus increased and decreased at 4, 8 dpi; 2, 4 dpi, respectively. Plasma MDA increased at 2 dpi, SOD, CAT, and T-AOC decreased at 2, 4, 8 dpi; 4, 8 dpi; and 4, 6, 8 dpi, respectively. These results suggest that rMDPV infection led to early intestinal barrier dysfunction, inflammation, ileac microbiota disruption, and oxidative stress.

## 1. Introduction

Parvovirus emerged tens to hundreds of millions of years ago, with wide host ranges [1,2]. Muscovy duck parvovirus (MDPV) belongs to the *dependent parvovirus* genus of *Parvoviridae*. MDPV was first reported in China in 1991 [3], which has also been reported in the USA [4,5]. The disease caused by MDPV is called “Muscovy three-week disease” because the clinical signs and symptoms caused by MDPV primarily occur in ducklings before the age of 3 weeks [3,6,7]. MDPV infects Muscovy ducks or mule ducks only, and the major clinical symptoms include developmental delays, weakened feet, panting, diarrhea, and death [3,4,6]. MDPV has multi-stage replication in organs [6,8,9,10], causing pericardial effusion, liver lesions, pancreatic necrosis, and other symptoms [7,11], posing a threat to the waterfowl industry.

The MDPV genome has two open reading frames (ORFs). ORF1 encodes the nonstructural protein (NSP), also known as regulatory proteins (REP), which are involved in the regulation of viral replication, and few studies have been conducted on the REP of MDPV. ORF1 produces REP1 and other REP based on the initial codon and selective splicing of mRNA [12]. ORF2 encodes three capsid proteins: VP1, VP2, and VP3 [13]. The initiation codon positions of the three genes are different, but the termination codon is the same. Among these genes, VP1 contains all amino acid sequences of *VP2* and *VP3* [14,15], and VP3 is the major capsid protein, accounting for 80% of the total capsid protein [14]. VP3 can effectively induce neutralizing antibodies against MDPV in Muscovy ducks [16]. The two ends of the MDPV genome have inverted terminal repeat (ITR), which is closely related to gene replication and transcription [17].

Goose parvovirus (GPV) and MDPV belong to the same genus of dependent parvoviruses [18], which can be collectively referred to as waterfowl parvovirus (WFPs) [19]. Both viruses have similar gene structures and patterns of gene expression [20,21]. The two parvoviruses outbreak simultaneously and frequently, leading to genetic recombination [22,23], placing a great burden on the waterfowl breeding industry. The rMDPV can cause diseases with higher mortality [24,25], which has been reported repeatedly in southern China [26,27,28,29]. The 1.1 kb region in the middle of the *VP3* gene of rMDPV contains the homologous sequence of GPV [28], which may cause GPV-specific intestinal embolism symptoms [11].

Parvovirus causes gastrointestinal diseases, particularly in birds [30]. Parvovirus replicates in the intestinal tissue first [19], leading to intestinal inflammation and an imbalance of intestinal flora [31]. Infection of WFPs is related to intestinal flora and the intestinal barrier [32,33,34]. Animal health is closely related to the intestinal microbiota [35]. Intestinal microorganisms exist in a symbiotic relationship with their hosts, which can affect the intestinal immune system [36]. The invasion of pathogens could disrupt the intestinal environment, inducing intestinal diseases and aggravating the symptoms [37]. Although they remove bacteria, cytokines also increase intestinal permeability and enhance virus-induced inflammation [38]. The decrease in antioxidant capacity caused by viral plasma infection can also aggravate the disease. To date, no detailed report on the intestinal pathogenesis of rMDPV infection in Muscovy ducklings has been found.

In this study, we obtained an MDPV strain from ducklings of Yunfu of the Guangdong Province, namely, the XMX strain. The ducklings had diarrhea and weakened feet before death, and no obvious lesions were found after dissection. The effects on the intestinal barrier, intestinal microbiota, and mucosal immune system of one-day-old Muscovy ducklings were investigated. The sequence analysis was conducted, and the XMX strain was identified as rMDPV. These results may provide information about the pathogenicity and evolution of WPFs.

## 2. Materials and Methods

### 2.1. Virus and Viral DNA Extraction

The livers and spleens from the dead ducks were collected, homogenized, and suspended in phosphate-buffered saline (PBS). After centrifugation, the supernatants were filtered through 0.22 μm filters and then extracted by AxyPrep Body Fluid Viral DNA/RNA MiniPrep Kit (AxyGen, Shenzhen, China) and stored at −80 °C. Then, they were subjected to PCR or RT-PCR for the detection of potential pathogens, including Goose parvovirus (GPV), duck Tembusu virus (DTMUV), egg drop syndrome virus (EDSV), duck hepatitis A virus type 1 (DHAV-1), Baiyangdian (BYD) virus, adenovirus, infectious bronchitis virus (IBV), H9N2 avian influenza viruses (AIV) and C. psittaci. The specific primer pairs of GPV [39] and others [40] were all the same as previous studies by our laboratory.

The virus was propagated in 12-day-old specific-pathogen-free embryonated Muscovy duck eggs (Wens, Yunfu, China), as described in previous studies [26]. The embryos were incubated at 37 °C for five to seven days and monitored daily. The virus titration with EID_50_ was determined in the same 12-day-old eggs by the Reed–Muench method [41].

### 2.2. Animal Experiments

Sixty-one-day-old Muscovy ducklings free of parvovirus-specific maternal antibodies (WENS, Yunfu, China) were randomly divided into the control group and infected group, with 30 ducklings in each group. All ducklings were housed and bred at a pathogen-free animal isolation facility at 25 °C–30 °C, provided with adequate commercial feed and water. According to a previous report [26], the ducklings in the infected group were infected intramuscularly with 10^6^EID_50_ of the XMX strain in the leg at 1 day of age, whereas the ducklings in the control group were injected with PBS in the same manner. The clinical signs of disease of all ducklings were measured daily. Six ducklings of each group were sampled after being euthanized by CO_2_ at 2, 4, 6, and 8 days post-infection (dpi).

### 2.3. Sample Collection

The plasma samples were collected in a pyrogen-free vacuum anticoagulant vessel via the cervical vein from ducklings at 2, 4, 6, and 8 dpi before euthanizing. The ileum of ducklings was obtained after being euthanized. Half of the samples were preserved in 4% formaldehyde for fixation, embedding in paraffin wax, section preparation, hematoxylin and eosin (H&E) staining, and histopathological analyses. The other half of the samples were snap-frozen in liquid nitrogen for the detection of viral load and gene expression. The intestinal contents were collected from the ileum of each duckling and then snap-frozen in liquid nitrogen for total microbial analysis.

### 2.4. Real-Time Quantitative PCR (qRT-PCR)

The primers F: 5′-TGTAGCTTGGCAGGACCCAG-3′; R: 5′-GGTGCCGATGGAGTGGGTAA-3′ were used for PCR amplification to generate standard plasmid ligated with pMD-19T vector and detect viral load. The plasmid concentration was measured and diluted into different concentrations for qRT-PCR, and the standard curve was calculated and obtained. The viral genome of ileac tissue was extracted by the AxyPrep Viral DNA/RNA MiniPrep Kit (AxyGen, China). The primers mentioned above were used for qRT-PCR, and the copy number of the XMX strain was calculated by standard curve.

RNA of ileac tissue was extracted by RNAiso Plus reagent (Takara, Beijing, China). According to the instructions, the cDNA was prepared by the HiFair™ II 1st Strand cDNA Synthesis Kit (Yasen, Ningbo, China). The expression of tight junctions (TJs) protein genes (*ZO-1*, *Occludin*, and *Claudin-1*) and immune-related factors (*IFN-α*, *IL-1β*, *IL-2*, *IL-6*, and *MHC-II*) in the ileum were detected by the 2^−∆∆CT^ method and normalized based on the expression level of the internal control GAPDH gene. The specific primers are shown in Table 1. The qRT-PCR was performed by CFX96 Real-Time Detection System (Bio-Rad, Hercules, CA, USA).

### 2.5. Assay of Intestinal Flora

Intestinal bacterial genomic DNA was extracted by the TIANamp Stool DNA Kit (TIANGEN, Beijing, China). DNA samples were quantified using a NanoDrop spectrophotometer (Thermo, Waltham, MA, USA), and the integrity of the DNA samples was tested via agarose gel electrophoresis. Sequencing libraries were generated using the NEBNext^®^ Ultra™ IIDNA Library Prep Kit. The library quality was evaluated on the Qubit (Thermo Scientific, Boston, MA, USA) and Agilent Bioanalyzer 2100 system, and the library was sequenced on an Illumina NovaSeq platform. The data were obtained by separation based on barcode and primer sequence, stitched by FLASH (V1.2.11) [42], and filtered chimeras by Vsearch (2.15.0) [43]. Denoising was performed with DADA2 in QIIME2 to obtain initial amplicon sequence variants (ASVs), and then ASVs with an abundance of less than 5 were filtered out [44]. Species annotation was performed on the ASV of each sample at different levels by classify-sklearn in QIIME2. Subsequently, α-diversity and β-diversity were analyzed by QIIME2.

### 2.6. Detection of Plasma Antioxidant Capacity

Superoxide dismutase (SOD), malondialdehyde (MDA), catalase (CAT), glutathione peroxidase (GSH-PX), and total antioxidant capacity (T-AOC) levels in plasma were determined on the basis of the instructions by the kit provided by Nanjing Jiancheng Bioengineering Institute.

### 2.7. Genome Sequencing and Analysis

Sequencing primer pairs were designed based on the complete genomic sequence of all MDPV published in GenBank (Table 2). The primers were used to amplify overlapping fragments by Prime STAR Max DNA polymerase (Takara, China). DNA fragments were gel-purified by the Gel Extraction Kit (Omega, Stamford, CT, USA), and then subjected to direct sequencing with amplification primers. The sequencing of ITR was based on the previous report [28]. The obtained sequences were combined to obtain the whole genome sequence using the Seq Man program of DNA star 5.0. Sequence alignment was performed by MEGA-X [45]. Nucleotide sequences of the *VP3* gene of XMX and strains in Table 3 were compared using Edit seq Program and translated into amino acid residues. The phylogenetic trees based on all WPFs genomic sequences from GenBank were generated by MEGA-X. RDP4 [46] and Simplot [47] were used to detect recombination events and recombination breakpoints under default settings based on all WPFs genomic sequences from GenBank. The data generated by Simplot were exported and presented by GraphPad Prism 8.0.

### 2.8. Statistics

Data analysis was conducted using SPSS 19.0, and the results were shown as mean and standard deviation. The Tukey method was adopted for multiple comparisons, the data were displayed as mean and standard deviation (SD), and *p* < 0.05 was considered statistically significant. GraphPad Prism 8.0 was used to create the artwork.

## 3. Results

### 3.1. XMX Infection Causes Intestinal Structure Injury

In total DNA and RNA, the GPV, DTMUV, EDSV, DHAV-1, BYD virus, adenovirus, IBV, H9N2 AIV and *C. psittaci* were all negative. The animal operation is shown in Figure 1A. Ileal viral load peaked at 2 dpi (Figure 1B). In the infected groups, mucosal injury and villus structural incompleteness were observed at 2 dpi, and seriously damaged mucosa and villus lost basic structure were observed at 4, 6, and 8 dpi. The structure of the mucosa was clear, and the villus was abundant in all the control groups (Figure 1C). The gene expression of *ZO-1*, *Occludin*, and *claudin-1* in the ileum is shown in Figure 1D. The *ZO-1* was significantly downregulated at 4 and 6 dpi (*p* < 0.05). The *Occludin* was significantly downregulated at 4, 6, and 8 dpi (*p* < 0.05). The *Claudin-1* was significantly downregulated at 2, 6, and 8 dpi.

### 3.2. XMX Infection Affects mRNA Expression of IFN-α, IL-1β, IL-2, and IL-6

The gene expression of *IFN-α*, *IL-1β*, *IL-2*, *IL-6*, and *MHC-II* in the ileum is shown in Figure 2. The *IFN-α* was upregulated significantly at 6 dpi (*p* < 0.05). The *IL-1β* was upregulated significantly at 8 dpi (*p* < 0.05). The *IL-2* was reduced significantly at 6 and 8 dpi (*p* < 0.05). The *IL-6* was upregulated significantly at 6 and 8 dpi (*p* < 0.05). The *MHC-II* had no significant difference at 2, 4, 6, and 8 dpi compared with the control groups (*p* < 0.05).

### 3.3. XMX Infection Affects the Diversity of Ileac Microbiota

A total of 5,588,953 raw reads were obtained from 48 samples with an average read length of 410 bp. After removing the primers, low-quality sequences, and filtering chimeras, we obtained 4,340,651 optimized sequences, which were used to profile the ileac microbiomes ASVs as proxies for bacterial species. The rarefaction curve was constructed by sequencing data volume and corresponding index value, which was used to describe the sample diversity within the group. In the present study, with the increase in sequencing depth, the observed ASVs tended to plateau, which indicated that the number of species in the ileac contents did not increase significantly (Figure 3A), showing that the sample was sufficient, and the results of further analysis were reliable. Non-metric multidimensional scaling (NMDS) analysis reflects the differences between and within groups through the distance between points. The samples with high community structure similarity tended to cluster. In this study, the points of all eight groups could be divided into two groups: the control group (C2, C4, C6, and C8) and the infected group (I2, I4, I6, and I8). The points of the four infected groups (I2, I4, I6, and I8) in the NMDS plot shared a single area (Figure 3B). In this study, the Ace and Chao1 index of I4 was significantly higher than that of C2, I2, C4, C6, C8, and I8 (*p* < 0.05), whereas I6 and I4 showed no significant difference (*p* < 0.05). The Shannon indexes in eight groups showed no significant difference (*p* < 0.05), whereas the Simpson index of I4 was significantly lower than that of C4, C6, I6, and C8 (*p* < 0.05, Figure 3C). The infection of XMX increased the richness of the ileac microbial community in ducklings at 4 dpi and returned to normal at 8 dpi.

### 3.4. XMX Infection Affects the Composition of Ileac Microbiota

At the phylum level, *Firmicutes* and *Proteobacteria* were the two most abundant bacteria (Figure 3D). *Firmicutes* in I2, I4 showed a significant decrease relative to C2, C4 (*p* < 0.05). However, *Proteobacteria* in I2, I4 showed a significant increase relative to C2, C4 (*p* < 0.05). At the genus level, the dominant genera were *Ochrobactrum*, *Enterococcus*, *Rothia*, *Streptococcus*, *Shigella*, *Ralstonia*, and *Clostridium* (Figure 3E). *Ochrobactrum* in I4, I8 showed a significant increase relative to C4, C8 (*p* < 0.05). *Enterococcus* in I2, I4 showed a significant decrease relative to C2, C4 (*p* < 0.05). *Streptococcus* in C8 was significantly higher than all groups (*p* < 0.05), and no significant difference was found in the other groups (*p* > 0.05). No significant difference in *Rosia* was found in all groups (*p* > 0.05).

### 3.5. XMX Infection Reduces Plasma Antioxidant Capacity

The plasma antioxidant capacity indicated by MDA, SOD, CAT, GSH-Px, and T-AOC is shown in Figure 4. The MDA of the infected group at 2 dpi was significantly higher than that of the control group (*p* < 0.05). The SOD of the infected groups at 2, 4, and 8 dpi was significantly lower than that of the control group (*p* < 0.05). The CAT of the infected group at 4 and 8 dpi was significantly lower than that of the control group (*p* < 0.05). The T-AOC of the infected group at 4, 6, and 8 dpi was significantly lower than that of the control group (*p* < 0.05). No significant differences in GSH-Px were observed between the infected and control groups at 2, 4, 6, and 8 dpi (*p* > 0.05). The above-mentioned results indicated that XMX infection could lead to a decrease in plasma antioxidant capacity, thereby resulting in oxidative stress.

### 3.6. Genomic Sequence and Analysis

The whole genomic sequence of the XMX strain was obtained and submitted to GenBank (MZ334491). The complete genome of the XMX strain encoding *REP* of 627 amino acids(aa) (nt 517–2400), *VP1* of 732 aa (nt 2419–4617), the sequences encoding *VP2* and *VP3* began at positions 2854 and 3013 nt, respectively. The *ITR* located at the 5′ and 3′ terminal ends of the genome was 386 nt long.

All sequences of the three phylogenetic trees of the complete genome, *REP*, and *VP1* were divided into GPV, MDPV and rMDPV. The XMX strain and all reported rMDPV (ZW, JH06, JH10, SAAS-SHNH, FJM3, and GDNX) were clustered in the same branch (Figure 5A–C). The phylogenetic tree of *ITR* can only be divided into two branches: GPV and MDPV (Figure 5D).

The alignment results of the VP3 amino acid residue sequence are shown in Figure 6. The amino acid residues (64–395 aa) in the middle segment of the VP3 protein were similar to GPV. The amino acid characteristics of the XMX strain were consistent with those of the reported rMDPV strains, which had special mutations at 263, 291, and 354 aa.

### 3.7. Recombination Analysis

The result of recombinant analysis by RDP 4.101 is shown in Figure 7A. The XMX strain and five reported rMDPV strains (ZW, JH06, JH10, SAAS-SHNH, and FJM8) underwent identical recombination events from the same two miner parents at 482–675 nt and 3181–4321 nt, respectively. The first miner parent with the highest homology was GPV SYG61v (KC996729), located upstream of *REP*, accounting for 9.29% of REP sequence length. The second miner parent with the highest homology was GPV SQ0412 (MF942876), which was located in the middle of *VP3*, accounting for 72.3% of the length of the *VP3* gene. In addition, the AH1401, GDNX, LH, and PT strains have different recombinant patterns (Figure 7A). The major parent of all the above-mentioned strains was MDPV YY (MF942876).

All WPF sequences were divided into three groups to verify the above-mentioned result, namely, GPV, MDPV and rMDPV, based on the three branches of the complete genome phylogenetic tree (Figure 1A). The result of SimPlot 3.5.1 is shown in Figure 7B. Compared with the XMX strain, the similarity of sequences in the rMDPV group was above 97%. The similarity curve of the MDPV group and GPV group intersected at four sites, indicating the two recombinant fragments in the XMX strain, which were obtained from the homologous fragment of GPV located at 321–561 nt and 3161–4321 nt, respectively, consistent with the analysis result of RDP4.101.

## 4. Discussion

Enterocytes of the small intestine are the primary target cells of parvovirus [19], and parvovirus infections can damage the intestinal structure [56,57], which was associated with intestinal diseases in avians [58,59,60]. In the present study, XMX replicates in the intestines of infected ducklings and significantly disrupts the intestinal villus structure. *ZO-1* binds to transmembrane and forms complete TJs [61]. *Occludin* and *Claudin-1* are crucial to TJs [62,63]. The changes in gene expression of *ZO-1*, *Occludin,* and *Claudin-1* in this study suggest the destruction of ileal TJs after XMX infection, which can be observed in other reports about viral infection [64,65,66,67,68,69]. The destruction of ileum villi after infection indicates that the intestinal absorption of nutrients is reduced, which may cause Muscovy duck dwarfism.

Given the intestinal inflammation, the intestinal proinflammatory cytokines significantly generated and affected intestinal epithelial barriers to increase leakiness [70,71]. *IFN-α* and *IL-1β* were relevant to host–pathogen interactions and intestinal inflammation [72,73,74,75,76]. *IL-6* can regulate crypt homeostasis and intestinal TJs [74,77]. The results of the present study suggested that inflammation occurred after XMX infection. In a previous study, the levels of IFN-α, IL-1β, and IL-6 in plasma were increased at 6 and 15 dpi in Cherry Valley ducklings infected with Duck original GPV (D-GPV) [32,78], consistent with the present study. The infection of porcine parvovirus and Human parvovirus B19 also increased *IL-6* expression [79,80]. *IL-2* is required for intestinal immunologic homeostasis [81]. The infections of avian influenza (AIV) and duck hepatitis A virus (DHAV) caused an increase in IL-2 in ducks [82,83,84], but the infection of duck plague virus (DPV) caused a decrease in IL-2 levels [85]. The production of IL-2 was significantly decreased in latently HIV-1-infected T cells [86,87], which contributed to the immune deregulation [88]. The decrease of IL-2 in this study suggested that the immune function of ducklings was damaged, which may be related to the pathogenesis of MDPV. *MHC-II* expression controls epithelial-cell remodeling following infection and mucosal immune responses [89,90]. D-GPV infection might alter the gene expression of *MHC-II* [78], and the intestinal structure of ducklings damaged by XMX infection will decrease intestinal absorption of nutrients and malnutrition and prevent the apparent normal increase in intestinal *MHC-II* expression in response to infection [90]. Thus, *MHC-II* expression was no different from the control groups. The duck cytokines remain poorly understood and require further verification, but we can leave it as a possibility.

The cause of enteric disease in poultry was complex and polymicrobial [19], and the damage of D-GPV to the intestinal barrier in Cherry Valley ducklings was closely related to the changes in intestinal flora [32,78]. The α-diversity index reflects the diversity of microorganisms, including the richness index (Chao1 and ACE) and the diversity index (Shannon and Simpson index). In the present study, the intestinal flora diversity of infected ducklings increased sharply at 4 dpi, then decreased, and reached the same level as the control group at 8 dpi, in line with other avian viral infection reports [32,91]. The increase in intestinal flora diversity in initial infection may be related to inflammation, whereas a subsequent decline may lead to disruption of the healthy host [92]. We hypothesized that the diversity of intestinal flora of ducklings might continue to decline after 8 dpi.

In the present study, in phyla, *Firmicutes* and *Proteobacteria* which were most abundant, decreased and increased, respectively, similar to the manifestation of preterm infants with necrotizing enterocolitis (NEC) in their feces [93]. *Enterococcus* and *Ochrobactrum* are most abundant in genera, belonging to *Firmicutes* and *Proteobacteria*, respectively, and the change in richness was similar to that. *Enterococcus* is Gram-positive, facultatively anaerobic oval cocci, which are present as natural colonizers of the gastrointestinal tract in most humans and animals, and they have been used safely as probiotics [94]. *Enterococcus* can improve broiler live performance [95] and gut microvillus [96]. *Enterococcus* might regulate inflammatory responses [97], and viral infection can reduce the number of intestinal *Enterococcus* in piglets [98]. After further analysis, the *Enterococcus* mentioned above is *Enterococcus cecorum* (EC). The EC can cause disease in broiler chickens [99] but does not cause spondylitis and femoral head necrosis in ducks [100]. *Ochrobactrum* is Gram-negative, aerobic, rod-shaped, non-pigmented, and motile opportunistic pathogens, which have caused many separate outbreaks [101]. The increase in *Ochrobactrum* was associated with NEC and ulcerative colitis exacerbated in humans [102,103]. In this study, the diversity of *Enterococcus* and *Ochrobactrum* was decreased and increased in the ileum after infection, which indicated a decline in the intestinal health of ducklings. Strategies to alter the intestinal microbiota might reduce disease severity [104]. A report showed that fecal microbiota transplantation in parvovirus-infected puppies was associated with a faster resolution of diarrhea [33]. Therefore, we should pay attention to the influence of shifts of intestinal flora after infection and morbidity, which might reduce the loss from the perspective of intestinal flora. Collectively, XMX infection in Muscovy ducklings resulted in a shift in the diversity and composition of ileac microbiota, which may contribute to the disease.

Oxygen has a deleterious effect on biomolecules in the form of free radical and reactive oxygen species (ROS) [105], and many viruses have been demonstrated to cause cell damage by generating ROS and changing redox homeostasis [106,107,108,109]. The body can defend itself against ROS by using enzymes such as SOD, CAT, and GSH-Px [110,111,112]. MDA is the principal product of polyunsaturated fatty acid peroxidation, which is considered a highly toxic molecule and a marker of lipid peroxidation [113]. In this study, we observed that XMX infection induced high ileal MDA and decreased CAT, SOD, and T-AOC activities, indicating the decreased antioxidant capacity. Innate immune cells are activated in all viral infections, causing ROS and prooxidant cytokines [114], and they are involved in the destruction of microbes, viruses, and infected cells [115]. Viral infections also enhance the production of oxidants and prevent the synthesis of CAT, SOD, and GSH-Px, resulting in the disruption of the redox balance [112], leading to a weak immune response [115], and promoting viral infection. Moreover, ROS interferes with the antigen presentation by innate immune cells, T cell polarization, and adaptive immune responses [116]. Furthermore, the immunosuppressive effects of ROS may facilitate viral infection and evolution [117]. We observed a decrease in plasma antioxidant capacity in ducklings infected with XMX, indicating the existence of oxidative stress and promoting viral infection.

In this study, the symptoms are also worth discussing. MDPV can cause intestinal disease, and ducklings are primarily presented with diarrhea [7,26,118] and intestinal embolism [11], but some MDPV infections do not show the symptoms mentioned above [4,5]. D-GPV is widely pathogenic to waterfowl, but its symptoms vary, and atrophic beaks and protruded tongues are not observed; thus, this finding is inconsistent with the previous report [119]. Given that the experiment was conducted on 8-day-old ducklings, the growth inhibition and atrophic beaks caused by MDPV infection cannot be observed. The duckling showed depression, foot fatigue, mouth breathing, and diarrhea, but no intestinal embolism was reported previously [11] after the challenge, which indicated that the pathogenicity of WFPs was diverse. To date, no multiple identical reports have confirmed a difference in pathogenicity between WPFs, further comparative experiments are needed.

The recombinant analysis showed that the beginning and ending breakpoints of recombinant fragments were consistent with those of rMDPV ZW [24], JH06, JH10 [28], SAAS-SHNH [25], and FJM3 [48]. The minor parent of the recombinant fragment of the *REP* is the vaccine strain GPV SYG61v (KC996729), which was consistent with other reports [24,25,28], has been used to protect goslings from GPV infection in China for 40 years [52,120], promoting co-infection and recombination. The minor parent of the *VP3* gene recombination fragment of the XMX strain and all rMDPV was GPV SQ0412 (MF942876), but previous reports suggested that many rMDPV had different sources, such as YZ99-6 [24], DY16 [28], and E [25]. Thus, we compared the recombinant fragment of the XMX strain (3181–4312 nt) with the homologous regions of the above strain and the results showed that the homology was 95.1%, 93.5%, 95.0%, and 93.7%. SQ0412 had the highest homology, indicating that the analysis of the present study was valid. The SQ0412 strain has not been reported yet, and it is related to commercial vaccines, according to the information from the GenBank website page.

The VP3 amino acid residues of the XMX strain were consistent with those of other MDPV at both ends (6–35 and 462–509 aa), and the middle part (64–395 aa) was consistent with GPV. The VP3 protein can induce humoral immunity [14,16], and may also be related to pathogenicity. Compared with the earliest rMDPV, namely, ZW and JH06 (collected in 2006), the newly isolated GDNX (collected in 2016) has changed in amino acid residue characteristics of VP3, which is the same as the XMX strain, such as 291 aa, indicated the evolving of MDPV.

At present, the reports on WPFs are primarily focused on molecular characteristics and few report on the pathogenicity. The differences in virulence, cytokine profiles, and intestinal microflora changes can be compared. In the present study, XMX infection of one-day-old Muscovy ducklings can cause severe damage to the ileum villus structure, affect the expression of intestinal TJs and ileum cytokines, alter the diversity and composition of ileum microflora, and cause oxidative stress, which has a negative influence on duckling health. In addition, the XMX strain belongs to rMDPV, and the characteristics of the XMX strain are consistent with other rMDPV. These results may provide information about the pathogenicity and evolution of WPFs.

## 5. Conclusions

MDPV XMX infection led to intestinal barrier dysfunction, inflammation, ileac microbiota disruption, and oxidative stress, which might be related to a variety of symptoms later in ducklings.

## Figures and Tables

**Figure 1 viruses-14-01471-f001:**
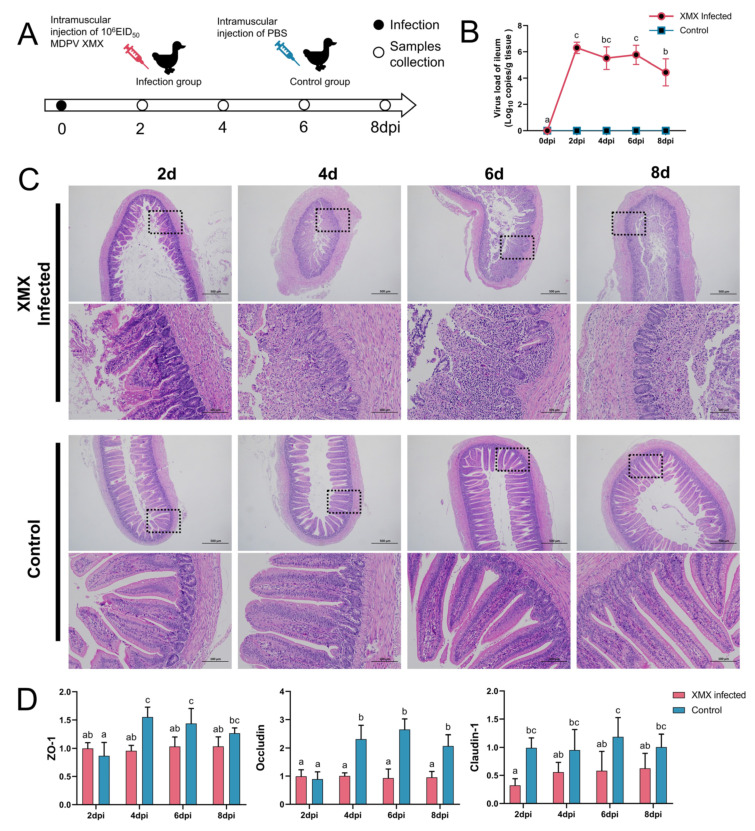
Influence of XMX infection on the ileal epithelial structure. (**A**) Schematic diagram of XMX infection and animal operations; (**B**) Viral load in the ileum; (**C**) Hematoxylin and eosin staining shows the ileal histological features in the XMX infection and control groups; (**D**) qRT-PCR was used to detect the gene expression of *ZO-1*, *Occludin* and *Claudin-1* in the ileum. Different letters indicate significant differences.

**Figure 2 viruses-14-01471-f002:**
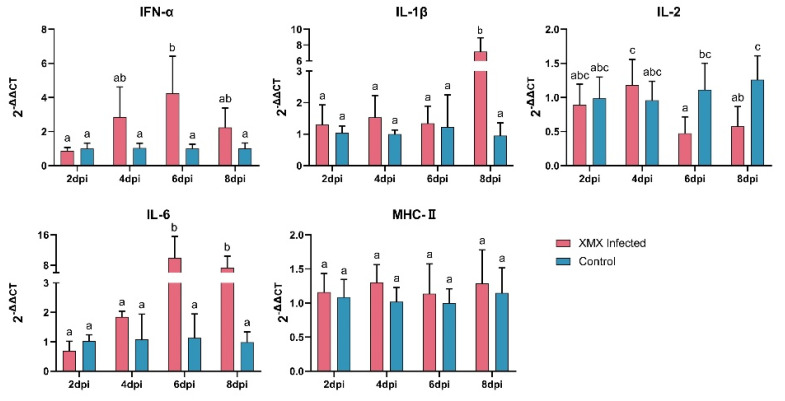
The mRNA expression of *IFN-α*, *IL-1β*, *IL-2*, *IL-6*, and *MHC-II* in the ileum after XMX infection, as found by qRT-PCR. Different letters indicate significant differences.

**Figure 3 viruses-14-01471-f003:**
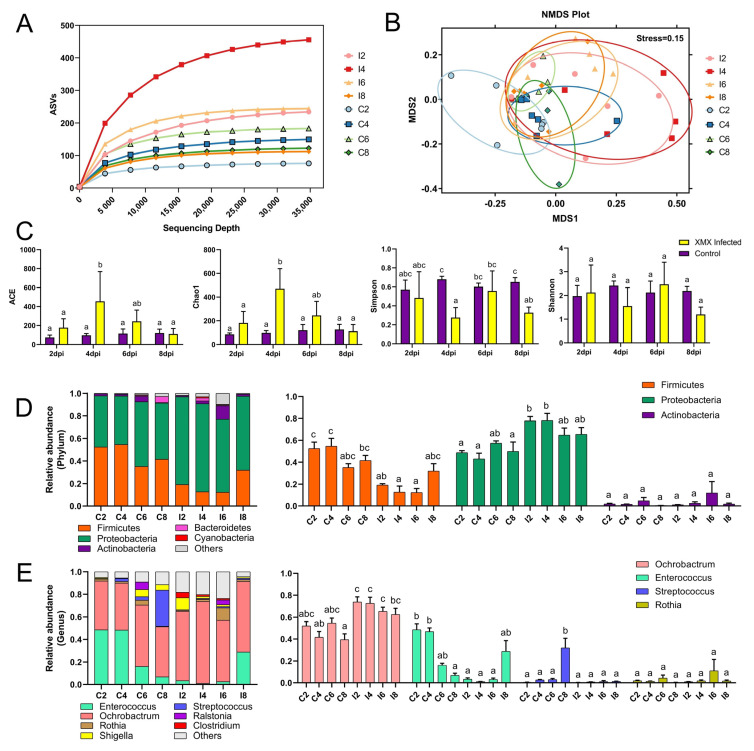
Effect of XMX infection on the diversity and component of ileal microbiota. (**A**) Rarefaction curve of the eight groups, and the horizontal axis represents the amount of sequencing data, the vertical axis represents the observed ASVs; (**B**) NMDS plot of the eight groups, each point in the plot represents a sample, the distance between points indicates the degree of difference, and the samples in the same group are represented by the same color. If the stress is less than 0.2, then NMDS can accurately reflect the degree of difference among samples; (**C**) α−Diversity of the eight groups; (**D**) Stacked bars represent the average relative abundance of the top 5 most abundant phyla within the ileum, “other” refers to the species outside the aforementioned top 5 phyla; (**E**) Stacked bars represent the average relative abundance of the top 7 most abundant genera within the ileum, “other” refers to the species outside the aforementioned top 7 genera. Different letters indicate significant differences.

**Figure 4 viruses-14-01471-f004:**
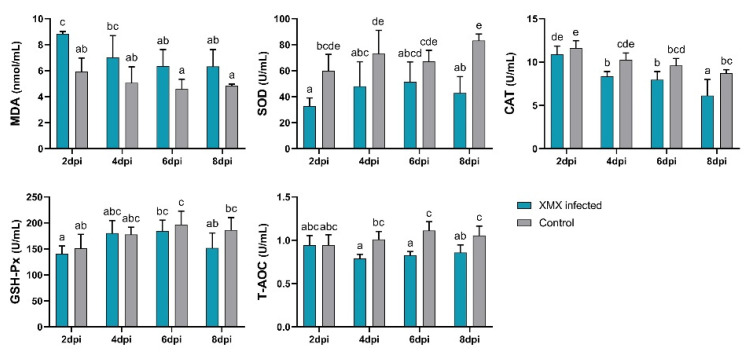
Plasma levels of SOD, MDA, GSH-Px, CAT, and T-AOC. Different letters indicate significant differences.

**Figure 5 viruses-14-01471-f005:**
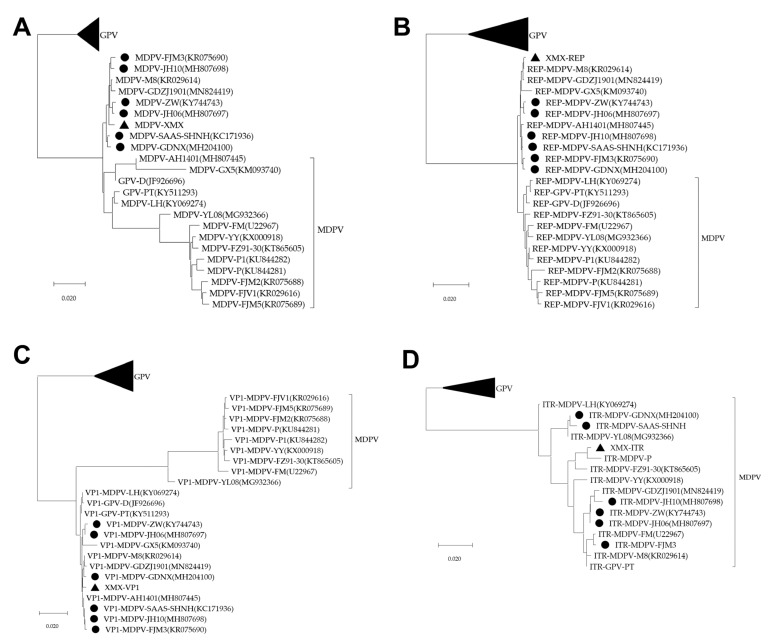
Phylogenetic analysis, based on complete MDPV and GPV genomic sequences on GenBank. The phylogenetic tree was constructed via the neighbor-joining method using the Maximum Composite Likelihood method and 1000 bootstrap replicates. Accession numbers of strains are indicated in parentheses. The black origin represents rMDPV; the triangle represents the XMX strain. (**A**) Phylogenetic tree of complete genomic sequence; (**B**) Phylogenetic tree of REP sequence; (**C**) Phylogenetic tree of VP1 sequence; (**D**) Phylogenetic tree of ITR sequence.

**Figure 6 viruses-14-01471-f006:**
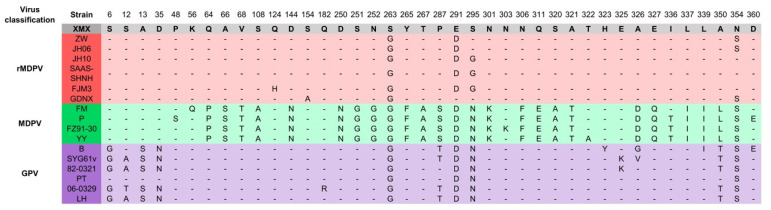
Alignment of amino acid residues of VP3 protein. Only the differentiated amino acid residues are shown.

**Figure 7 viruses-14-01471-f007:**
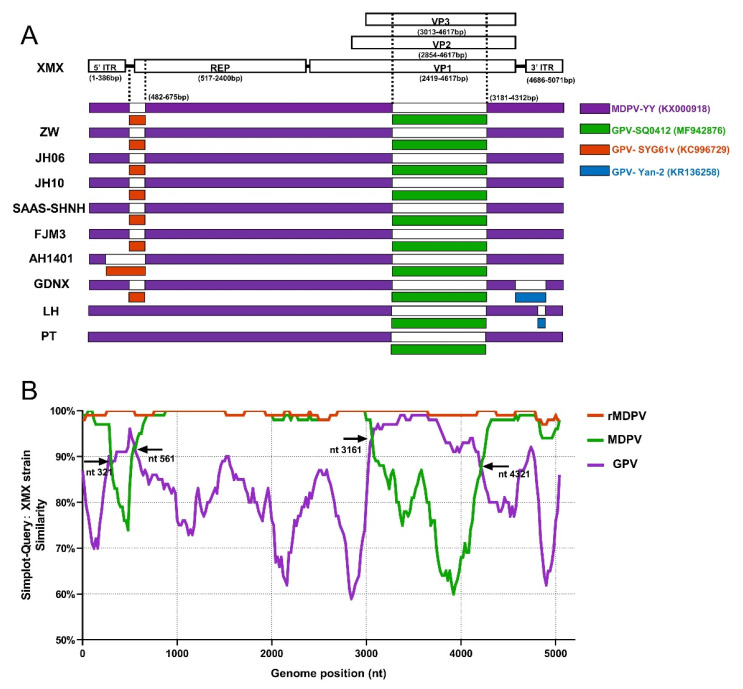
Recombination analysis based on complete MDPV and GPV genomic sequences on GenBank. (**A**) Recombination analysis of WPFs generated by RDP 4.4.3; (**B**) Recombination analysis by Simplot 3.5.1 presented by GraphPad Prism 8.0.

**Table 1 viruses-14-01471-t001:** Sequence of primers used in qRT-PCR to detect gene expression.

Gene	Forward Primer (5′–3′)	Reverse Primer (5′–3′)	Accession
ZO-1	TTCGGGAAGCTGGGTTTCTC	CCAGCGTCTCTTGGTTCACT	101791291
Occludin	GGGACTACGGCTACGGTTTG	CACCAGCAGCCCCAACTG	XM 013109403
Claudin-1	GATACTCCTGGGCCTCGTTG	CGAGCCACTCTGTTGCCATA	101797741
IFN-α	CACCTCTTCGACACCCTCAG	AGGTGGTGGATGTGGTGC	JF894229.1
IL-1β	TCACAGTCCTTCGACATCTTCG	CCTCACTTTCTGGCTGGATGAG	DQ393268
IL-2	GATGAGAACGTATCTAGTGTTCGG	CAGCTCTCGGCGAAATTCAG	AY193713.1
IL-6	CCAAGGTGACGGAGGAAGAC	GTTGCCAGATGCTTTGTGCT	101798321
MHC-II	GAGACCAAGGGGTTCTTCCA	TGCCGGTTGTAGATGTCTCTC	DQ490138.1
GAPDH	GCCACACAGAAGACAGTGGA	GTCAGGTCCACGACAGAGAC	GU564233.1

**Table 2 viruses-14-01471-t002:** Primer pairs designed for the amplification of complete genome of XMX strain.

Primer Names	Primer Sequences (5′–3′)	Primer Positions	Fragment Sizes (nt)
ITR-1	CTCATTGGAGGGTTCGTTCG	1–20; 5052–5071	187
ITR-2	GCCCGATCAGCCTTGACAAC	168–189; 4883–4904	
ITR-3	GCGCATGCGCCCGATCTGCCATGA	186–209; 4863–4886	532
ITR-4	GATTTTGTCTGCCAGAGTAACC	696–717	
1-F	CCCCATGGTTACTCTGGCAGACAAAA	690–715	927
1-R	GGGAAGTTCTCATTAGTCCAGT	1595–1616	
2-F	CCACCGGAAAGACCAACAT	1532–1550	935
2-R	GGCTGCAGTCTCATACCAGTCTT	2444–2466	
3-F	GCCTGGAGTGTGAAAGAGCTAATT	2237–2260	867
3-R	GGGAATCGCAATGCCAATT	3085–3103	
4-F	GAACCTGTGGCAGCACCTAACAT	2992–3014	1670
4-R	GCGCGCCAGGAAATGGTTTAT	4642–4662	
ITR-5	CCAAACCTGGGAGGTTTTGG	4303–4322	584
ITR-3	GCGCATGCGCCCGATCTGCCATGA	186–209; 4863–4886	

**Table 3 viruses-14-01471-t003:** Representative subsets of rMDPV, MDPV and GPV.

Virus Classification	Strain	Collection Year	GenBank Accession ID	Reference
rMDPV	ZW	2006	KY744743	[24]
	JH06	2006	MH807697	[28]
	JH10	2010	MH807698	[28]
	SAAS-SHNH	2012	KC171936	[25]
	FJM3	2013	KR075690	[48]
	GDNX	2016	MH204100	[22]
MDPV	FM	Unpublished	U22967	[49]
	P	1988	JF926697	-
	FZ91-30	1991	KT865605	[17]
	YY	2000	KX000918	[50]
GPV	B	Unpublished	U25749	[51]
	SYG61v	1961	KC996729	[52]
	82-0321	1982	EU583390	[53]
	PT	1997	JF926695	[54]
	06-0329	2006	EU583391	[53]
	LH	2012	KM272560	[55]

## Data Availability

The whole genomic sequence of the XMX strain was submitted to GenBank (MZ334491).

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
