# Peer review of "Recombinant Muscovy Duck Parvovirus Led to Ileac Damage in Muscovy Ducklings"

_viruses, 2022, doi:10.3390/v14071471_

Round 1
Reviewer 1 Report
I am very glad to find all the questions I have raised before were responded effectively in the new manuscript, and I think this mansuscript is ready for publication.
Author Response
We are glad to hear that, and we would like to thank this respected reviewer for approval. We hope everything goes well with you!
Reviewer 2 Report
Dear Authors,
resubmitting your paper allowed you to further improve it.
The manuscript is very well structured and the reading is fluent. I would like more space and visibility to be given to images, tables, graphs, and figures, since they are very well done and are very demonstrative of the results obtained, representing an essential component.
Therefore, in my opinion, with minor revisions, the manuscript will be ready for publication and will certainly meet the consent of scholars and readers.
Here are some of my comments and suggestions:
- Please remove from the keywords "Muscovy duck parvovirus" as it is already present in the Title;
- check the References section by following the template and the "Instructions for Authors" of the Journal.
Author Response
We are glad to hear that! Thank you very much for your approval of images, Tables, Graphs, figures. We hope everything goes well with you!
Q1: Please remove from the keywords "Muscovy duck parvovirus" as it is already present in the Title.
A1: Thank you for your careful reading and kind reminder, the keywords "Muscovy duck parvovirus" have been deleted. (Line 36)
Q2: Check the References section by following the template and the "Instructions for Authors" of the Journal.
A2: We would like to thank you for your comments. We have checked and revised the Reference section of the manuscript according to "Instructions for Authors", by the Style named “MDPI ACS Journals” of Endnote as requited. We also used manual modification. (Line 560, 569, 578, 605, 651, 679, 725, 736, 743)
This manuscript is a resubmission of an earlier submission. The following is a list of the peer review reports and author responses from that submission.
Round 1
Reviewer 1 Report
Dear Authors,
you have conducted a very interesting study and presented it very well. The manuscript is organized in a simple and clear way but then it has been enriched with figures, graphics, and images useful for understanding the results.
Therefore, in my opinion, I believe that your manuscript is already ready for publication in this form.
Reviewer 2 Report
1. Page 85: There is no description of whether the duck isolated virus has lesions or not, add please.
2. Page 94: Please describe the basis for using 12-day-old duck embryos to proliferate virus, and note reference.
3. Page 103: What is the basis for determining the inoculation route and virus dose in duck challenge test.
4. Page 358: The Enterococcus mentioned in this paper is Enterococcus faecalis or Enterococcus faecium? please specify.